# Risk Factors Associated with Corneal Nerve Fiber Length Reduction in Patients with Type 2 Diabetes

**DOI:** 10.3390/jcm14238411

**Published:** 2025-11-27

**Authors:** Lidia Ladea, Christiana M. D. Dragosloveanu, Ruxandra Coroleuca, Iulian Brezean, Eduard L. Catrina, Dana E. Nedelcu, Mihaela E. Vilcu, Cristian V. Toma, Adrian I. Georgevici, Valentin Dinu

**Affiliations:** 1Faculty of Medicine, Carol Davila University of Medicine and Pharmacy, 020021 Bucharest, Romania; lidia.ladea@drd.umfcd.ro (L.L.); christianacelea@gmail.com (C.M.D.D.); ruxandra.coroleuca@umfcd.ro (R.C.); dribrezean@yahoo.com (I.B.); eduard.catrina@umfcd.ro (E.L.C.); 2Clinical Emergency Institute of Ophthalmology Prof. Dr. M. Olteanu, 010464 Bucharest, Romania; 3Ophthalmology Department, Bucharest Emergency University Hospital, 050098 Bucharest, Romania; 4General Surgery Department, Cantacuzino Clinical Hospital, 011437 Bucharest, Romania; mihaela.vilcu@umfcd.ro; 5Ponderas Academic Hospital, 014142 Bucharest, Romania; nedelcudana68@gmail.com; 6Department of Urology, Carol Davila University of Medicine and Pharmacy, 020021 Bucharest, Romania; cristian.toma@umfcd.ro; 7Department of Urology, Clinical Hospital Prof. Dr. Burghele, 010464 Bucharest, Romania; 8Katholisches Klinikum, Ruhr University Bochum, 44791 Bochum, Germany

**Keywords:** diabetic neuropathy, corneal sensitivity, corneal nerves, corneal nerve fiber, CNFL, carotid plaque, intima-media thickness, HbA1c

## Abstract

**Background**: Diabetic neuropathy affects almost half of diabetic patients, yet the relative contributions of metabolic, vascular and clinical factors remain controversial. We aimed to investigate which risk factors are more associated with reduced corneal nerve fiber length (CNFL). **Methods**: This is a cross-sectional study of 30 patients with type 2 diabetes. We assessed metabolic parameters (HbA1c, lipids), vascular measurements (Doppler ultrasonography of carotid and ophthalmic arteries, central vessel density measured by optical coherence tomography angiography), and corneal epithelial thickness. We explored the data using network analysis, then applied penalized mixed-effect regression (in which β represents the standardized coefficients with mean 0 and unit standard deviation), followed by generalized additive models and polynomial transformations. **Results**: Penalized regression identified vascular parameters as dominant predictors: carotid plaques (β = −0.609) and intima-media thickness (β = −0.574) showed the strongest associations with CNFL. Traditional metabolic markers including HbA1c failed to meet selection thresholds. Bifurcation velocity (β = −0.313) and corneal sensitivity measures (β = 0.278–0.135) were also significant. The non-linear modeling showed complex vascular–structural interactions. **Conclusions**: Vascular compromise, particularly carotid disease, had the highest association with CNFL in our cohort. Thus, our study reports a higher effect of vascular parameters than HbA1c in patients with a longer history of diabetes. This may reflect the progression of diabetic complications, where initial metabolic insults are followed by vascular pathology as the primary driver of end-organ damage. Our findings highlight the need for carotid artery screening in diabetic patients for a better estimation of the neuropathy risk.

## 1. Introduction

Diabetic peripheral neuropathy (DPN) represents a major complication affecting up to one half of individuals with diabetes, leading to foot ulcers, amputations, and reduced quality of life [1,2,3]. In most cases, diabetic neuropathy presents as a symmetric polyneuropathy, affecting the sensory, motor, and autonomic components of the peripheral nervous system to varying degrees. The economic burden exceeds USD ten billion annually in the United States alone [4], with substantial costs for healthcare systems managing complex diabetes complications [5]. Current diagnostic methods like monofilament testing, vibration perception, and symptom questionnaires detect neuropathy only after substantial nerve damage has occurred. This diagnostic delay results in missed opportunities for early intervention, as routine clinical practice lacks sensitive tools for early neuropathy detection. Corneal confocal microscopy (CCM) has emerged as a highly sensitive method for detecting early small-fiber neuropathy [6]. The cornea, being the most densely innervated tissue, provides an accessible window to assess peripheral nerve health. Corneal nerve fiber length (CNFL), measured in mm/mm^2^, offers objective, quantifiable assessment of nerve damage with excellent reproducibility (Intraclass Correlation Coefficient = 0.87–0.94) [7]. Studies demonstrate CNFL reduction precedes clinical neuropathy [8]. Normal CNFL ranges from 19 to 25 mm/mm^2^, with values below 12 mm/mm^2^ indicating clinically significant neuropathy [7,9]. CCM parameters show strong correlations with sudomotor function [10]. Diagnostic accuracy studies report sensitivity of 85% and specificity of 84% for early DPN detection using a CNFL threshold of ≤14.0 mm/mm^2^ [11]. Recent advances in automated CCM analysis using artificial intelligence have significantly reduced operator dependency and improved reproducibility [12]. However, CCM requires expensive equipment and specialized expertise, limiting its availability. Despite the clear need for accessible early detection methods, no current tools integrate vascular parameters to quantify objective nerve damage. Previous attempts using single biomarkers showed limited accuracy, while complex machine learning models lacked interpretability and reproducibility for clinical use. This represents a critical gap, as routine diabetes care already includes vascular assessments that could potentially predict nerve health. Growing evidence supports the vascular hypothesis of diabetic neuropathy, wherein microvascular dysfunction drives nerve ischemia and subsequent degeneration [13,14]. The critical role of endothelial dysfunction in diabetic complications has been well-established, with impaired microcirculation directly affecting wound healing and tissue viability [15]. Moreover, the pathophysiology of critical limb ischemia demonstrates how vascular insufficiency creates a cascade of metabolic derangements that compromise neural tissue survival [16]. Atherosclerotic changes in large vessels (carotid arteries) and impaired perfusion in smaller vessels (ophthalmic artery) contribute to nerve damage across multiple end organs. Common carotid artery intima-media thickness (CCA IMT) correlates with neuropathy severity, while ophthalmic artery hemodynamics reflect local perfusion affecting corneal nerves. Recent studies demonstrate that increased CCA IMT is independently associated with reduced retinal vessel density and enlarged foveal avascular zone, confirming the link between macrovascular disease and microvascular compromise [17]. The Fremantle Diabetes Study Phase II provided compelling evidence that carotid disease parameters significantly correlate with retinal optical coherence tomography angiography (OCT-A) metrics, with IMT showing inverse relationships with deep capillary plexus vessel density [17]. Central vessel density (CVD), measured by optical coherence tomography angiography, provides a quantitative assessment of retinal microcirculation that correlates with systemic microvascular health [17,18]. Additionally, corneal epithelial thickness may indicate tissue resilience against diabetic damage. These parameters are measured during routine diabetes complication screening, presenting an opportunity for integrated assessment. The interdependency between vasculopathy and neuropathy involves complex non-linear interactions that simple linear models cannot capture. Polynomial transformations can reveal inflection points, saturation effects, and synergistic interactions between variables. Advanced non-linear modeling techniques could reveal hidden relationships between vascular parameters and nerve damage, potentially identifying critical thresholds and interaction effects that determine progression from subclinical to overt neuropathy. We hypothesized that vascular parameters would emerge as stronger predictors of CNFL than traditional metabolic markers in patients with diagnosed diabetes. Our primary objective was to use data-driven variable selection to identify the dominant predictors of nerve damage and to develop a regression model with R^2^ > 0.55. For this analysis, we used the patient’s worse eye (shorter CNFL) as the outcome, investigating associations between demographic factors, laboratory values, Doppler sonography findings, and ophthalmologic examinations with CNFL outcomes.

## 2. Materials and Methods

### 2.1. Study Design

This prospective observational study was conducted at the Clinical Emergency Institute of Ophthalmology Prof. Dr. M. Olteanu, Bucharest, Romania, and the Diapedis Clinic in Bucharest, Romania, between September 2023 and September 2024. The protocol, which adhered to the tenets of the Declaration of Helsinki (2013 revision), received approval from the local Institutional Review Boards. All participants provided written informed consent prior to enrollment. Inclusion criteria required participants to have a diagnosis of type 2 diabetes according to American Diabetes Association criteria [19], a disease duration greater than five years, a best-corrected visual acuity of 0.1 or better on the Snellen chart, and the ability to cooperate with all imaging assessments. Conversely, exclusion criteria included type 1 diabetes, current use of contact lenses, history of herpetic keratitis or other conditions known to affect corneal innervation, previous corneal or vitreoretinal surgery or significant pathology, and the presence of severe peripheral vascular disease that would preclude ultrasound examination.

### 2.2. Primary Aim

The primary aim of this study was to develop a regression model that estimates the maximum diabetic damage reflected in corneal nerve fiber length. Therefore, the analysis deliberately utilized data from the “worst eye”, defined by the lowest corneal nerve fiber length value, and its corresponding ipsilateral variables.

### 2.3. Sample Size

An a priori power analysis for a regression model indicated a minimum required sample size of 27 participants, assuming a medium-to-large effect size (expected R^2^ > 0.55), statistical power, and a defined significance level. To account for an anticipated dropout rate of approximately 10%, the target enrollment was set to 30 patients.

### 2.4. Clinical and Laboratory Assessments

Demographic data, diabetes duration, medications, blood pressure, lipid profile, smoking status, and history of myocardial infarction were collected. Each patient provided a blood sample for the analysis of metabolic and renal parameters, including HbA1c, high-density lipoprotein, low-density lipoprotein, total cholesterol, triglycerides, urea, creatinine, albumin/creatinine ratio, and estimated glomerular filtration rate. Peripheral neuropathy status was assessed using the Michigan Neuropathy Screening Instrument and SudoScan test (Impeto Medical, Paris, France).

### 2.5. Vascular Assessments

All Doppler ultrasonography examinations were performed by the same experienced operator with an ACUSON Redwood system (ASC3 Siemens Healthineers, Germany) after the patient had been at physical rest for at least 15 min prior to the examination. For carotid artery assessment, a high-resolution B-mode ultrasound with a 10 MHz linear probe was used to measure the common carotid artery intima-media thickness bilaterally, 1 cm proximal to the bifurcation on the far wall, according to established protocols. Three measurements were averaged per side, with the maximum value used for analysis. The common carotid artery caliber, resistivity index, and peak systolic velocity were also bilaterally recorded at the proximal and bifurcation sites. Plaques were defined as focal wall thickening greater than 1.5 mm or more than 50% of the adjacent intima-media thickness, and the total plaque count was recorded. For ophthalmic artery assessment, a 7.5 MHz probe placed on the closed eyelid was used to measure the peak systolic (PSV) and end-diastolic (EDV) velocities. The resistance index was calculated as (PSV-EDV)/PSV. Measurements were performed bilaterally, averaging three cardiac cycles per eye; however, for the final analysis, results corresponding to the eye with worse corneal nerve fiber length were used.

### 2.6. Corneal and Ophthalmologic Evaluation

The corneal evaluation began with esthesiometry, where corneal sensitivity was clinically assessed using a Cochet-Bonnet esthesiometer (Luneau, Prunay le Gillon, France) in the central cornea and all four corneal quadrants. Subsequently, an MS-39 anterior segment optical coherence tomography system (SCHWIND-CSO, Scandicci, Italy) was used to generate central corneal thickness assessment and epithelial thickness maps over the central 9 mm. The minimum epithelial thickness was recorded from the same eye used for the nerve fiber analysis. Following the application of topical anesthesia, the sub-basal nerve plexus was imaged using in vivo confocal microscopy (Heidelberg Retina Tomograph III with Rostock Cornea Module, Heidelberg Engineering GmbH, Heidelberg, Germany). Three non-overlapping, high-quality images were selected from the central cornea of each eye and analyzed with automated software (Corneal Nerve Fiber Analyser V.2, ACCMetrics, University of Manchester, Manchester, UK) to quantify corneal nerve fibre length (CNFL; total length of main nerves and branches in mm/mm^2^), corneal nerve branch density (CNBD; number of main branches per mm^2^), corneal nerve fiber area (CNFA), corneal nerve fractal dimension (CNFrD), and corneal nerve fibre density (CNFD; number of main nerves per mm^2^) [7,12]. A comprehensive ophthalmologic examination concluded with a dilated fundus examination (DFE) and stereo photographs of 7 standard fields around the macula and optic disc to assess and grade the clinical stage of diabetic retinopathy.

### 2.7. Statistical Analysis

The correlations between numerical variables were examined using three tests: Pearson, Kendall (non-parametric, analog to Spearman) and non-linear distance correlation. The associations between numerical and categorical variables was determined using Kruskall’s test, whereas for those between categorical variables we used the Chi^2^ Test. Significance was set at *p* < 0.05. Continuous variables are presented as mean ± SD or median (IQR) based on distribution. All analyses were performed used R version 4.5.1. Feature importance was performed using ensembles of decision trees supporting interactions based on the Boruta algorithm [20]. Each variable was tested as both predictor and target across 50 iterations, creating a complete interaction network. Boruta identifies both linear and non-linear multivariate interactions. Boruta score is the percent of decision trees which, across multiple iterations, have found a variable more informative for the given outcome than its permutated counterpart. We performed multilevel regression with L1 penalization to account for our hierarchical data structure in which eyes are nested within patients. This approach, which has been validated for obtaining reliable R^2^ values from generalized linear mixed models [21], incorporates patient effects and within-patient correlation while performing variable selection. The methodology for high-dimensional mixed-effect models with *ℓ*-penalization has been well-established [22], allowing us to jointly optimize fixed and cluster (random) effects while handling multicollinearity through shrinkage. The glmmLasso model [23] is specified as$${\mathrm{CNFL}}_{ij}=\beta_{0}+\sum_{k=1}^{p}\beta_{k}X_{ijk}+u_{i}+\epsilon_{ij}$$
where ${\mathrm{CNFL}}_{ij}$ is the corneal nerve fiber length for eye *j* of patient *i*; $X_{ijk}$ represents standardized predictors; $\beta_{k}$ represents fixed effects (subject to L1 penalty); $u_{i}∼N\left(0,\sigma_{u}^{2}\right)$ represents patient-specific random intercepts; $\epsilon_{ij}∼N\left(0,\sigma_{\epsilon }^{2}\right)$ represents residual errors. This data-driven approach allows the model to select the most informative predictors while controlling for overfitting. All pairwise interactions between numeric variables were tested using distance correlation (DCOR). Interactions exceeding dcor = 0.5 were further explored. We used non-linear regressions to capture complex relationships between vascular parameters and CNFL, namely generalized additive models with tensor product smooths [24]. This approach allowed visualization of non-linear interaction surfaces. We aimed also to investigate the best polynomial regression from all interactions.

### 2.8. Normative Benchmarking

Our study has no control healthy group; hence, we performed normative benchmarking against established reference populations. CNFL was compared to age- and sex-stratified values from Tavakoli et al. (n = 343 patients) [25], calculating z-scores as z=(observed−expected)/SD. CCA IMT was compared to population references, with pathological thickness defined as ≥0.9 mm per ESC/ESH guidelines [26].

## 3. Results

### 3.1. Patient Characteristics

Data from the thirty recruited patients was fully examined; only one patient had incomplete epithelial thickness data. We start by describing the categorical variables: in our cohort, we had 17 men and 13 women (n.s.); 16 were smokers (n.s.); half of them had a history of myocardial infarction (n.s.); 25 had arterial hypertension (*p* < 0.01); 28 of them were on oral antidiabetic drugs (*p* < 0.01); and 20 received insulin (*p* = 0.1). *p*-values were obtained using the binomial test. Table A1 shows pairwise correlations between study variables with at least one *p*-value ≤ 0.1, using linear (Pearson), monotonic (Kendall), and non-linear (distance) relationships. Table 1 summarizes the numeric variables for the study participants. Our study group consisted of middle-aged to older adults aged 59.5 [55–68] years (median [perc. 25–perc. 75]), with long-standing type 2 diabetes (10 [6.3–16.8] years). For bilateral measurements, the values presented are those corresponding to the ‘worst eye’, which is defined as the one with the shortest CNFL. The median CNFL from both eyes in our cohort was 11.88 ([9.98–13.51], range 7.47–17.77), whereas when we chose the worst eye per patient (10.71 [8.93–12.61], range 5.99–16.59). CNFL in all participants was below the normal range (19–25 mm/mm^2^) (7.9). Twenty patients had CNFL < 12 mm/mm^2^. For consistency, in case of bilateral measurements, we considered the worst eye.

### 3.2. Comparison with External Control Healthy Cohorts

Our data shows severe neuropathy with median CNFL z-score −1.91 (95% CI [−2.79, −0.86]), when compared to age-sex matched controls [25]. Two thirds of our patients had severe neuropathy (<11.8 mm/mm^2^). However, the health status of the carotid arteries was less pathological than the neuropathy: the median CCA IMT z-score was +1.08 (95% CI: [−0.95, 5.99]); here, only 16.7% of our patients reached pathological thresholds (>0.9 mm). The results indicate differential vulnerability of the carotid arteries compared to the nerve fibers to diabetic injury.

### 3.3. Correlations with CNFL

We assessed correlations between CNFL using three complementary methods: Pearson, Kendall, and distance correlation; significant ones are presented in Table 2.

Corneal nerve morphological parameters (fractal dimension, branch density, and fiber area) showed strong positive correlations with CNFL (r = 0.59–0.60). Corneal nerve fiber area (CNFA) and fractal dimension (CNFD) showed moderate positive correlation. The intima-media thickness of the common carotid artery (CCA IMT) demonstrated a significant negative correlation (r = −0.47) with CNFL. The linear correlation coefficient was r = −0.47 while the non-linear distance correlation was dcor = 0.54. The minimum epithelial thickness showed a moderate negative correlation (r = −0.40, *p* = 0.029) with CNFL.

### 3.4. Multivariate Association Network

The network of multivariate associations is presented in Figure 1. CNFL showed associations with multiple variables from the common carotid artery Doppler measurements (resistivity index, proximal peak systolic velocity, and intima-media thickness). Epithelial thickness, age, and arterial hypertension clustered together in the network.

The central vessel density showed connections to both metabolic parameters (cholesterol) and corneal nerve fiber width, which in turn was associated with the resistivity index in the ophthalmic artery.

### 3.5. Linear Predictors

The multilevel regression identified six predictors from 28 candidates. Sensitivity analyses showed that five predictors (CCA plaque, CCA IMT, CCA bifurcation velocity, central and temporal sensitivity) were selected across all analyses, while nasal sensitivity was selected in some but not all analyses (Table 3).

Carotid disease markers demonstrated the strongest associations, with presence of plaques (β = −0.609) and intima-media thickness (β = −0.574) showing the largest standardized coefficients. Carotid bifurcation blood flow velocity (β = −0.313) provided additional vascular contribution. Corneal sensitivity measures showed positive associations with CNFL: central (β = 0.278), temporal (β = 0.195), and nasal (β = 0.135) (Figure 2).

### 3.6. Non-Linear Interactions

Using distance correlation, we found variables which have a multiplicative (synergetic) association with CNFL. We explore these pairs of variables using generalized additive models (GAM). The GAM surfaces revealed patterns of vascular interactions: (i) CCA IMT × age (R = 0.64), showing compounding effects of aging and atherosclerosis; (ii) OA PSV × CCA IMT (R = 0.58), multiple-vessel interaction; (iii) central vessel density × CCA PSV bifurcation (R = 0.63), microvascular–macrovascular coupling; and (iv) central vessel density × OA PSV (R = 0.67), indicating a direct retinal perfusion relationship. Finally, we identified the polynomial regression with the highest R^2^ for the estimation of CNFL (Figure 3), which is given by the interaction between two standardized (z-scored) variables, namely the central vessel density (%) and the minimum epithelial thickness (μm):$${\mathrm{CNFL}}_{estimated}=0.54\times \left[CVD^{3}-0.51\times CVD^{2}-1.07\times CVD-{\mathrm{Epithelial}}_{min}\right]$$

## 4. Discussion

In our cohort, all participants already had significant corneal nerve damage before being enrolled in the study. Their median CNFL was well below the normal range, with two-thirds of our patients falling into the clinically significant neuropathy category < 12 mm/mm^2^ [27]. CNFL had significant positive correlations with other microscopic nerve measurements (like branch density, shown in Figure 2). More importantly, we observed a significant inverse correlation (−0.47) between CNFL and the thickness of the carotid artery wall (CCA IMT). Through network analysis (Figure 1) we observed that CNFL has multivariate interactions with common carotid artery (CCA IMT, CCA RI, CCA PSV), age, arterial hypertension, albumin/creatinine ratio and, unsurprisingly, other microscopic nerve measurements. Notably, HbA1c was not in the main interaction cluster of CNFL. Overall, Figure 1 provides an exploratory perspective of the interaction network of diabetic pathophysiology and reinforces that vascular and nerve health are interconnected. Using penalized regression, we found that vascular parameters are significant predictors of CNFL. Among 28 candidate variables, carotid plaques (β = −0.609) and intima-media thickness (β = −0.574) emerged as the strongest determinants of CNFL, with carotid blood flow velocity contributing to a lesser extent. Consistent with the network analysis, HbA1c was not selected as predictive for CNFL. This suggests that in patients with long-standing diabetes, current blood sugar control (HbA1c) is less associated with nerve damage than the health of large blood vessels. Thus, the accumulated vascular injury supersedes glycemic status as the proximate determinant of neural integrity. This interpretation challenges the metabolic-centric paradigm of diabetic neuropathy [2,28]. Nonetheless, we remind that our cohort comprises patients with a median diabetes duration of 10 years. Multiple mechanisms support vascular dominance in diabetic neuropathy pathogenesis. For example, basement membrane thickening and altered rheology further compromise neural perfusion [14]. The Fremantle Diabetes Study demonstrated strong correlations between carotid disease parameters and retinal microvascular changes [17], confirming parallel vascular injury across end organs. These alterations create oxidative stress and chronic hypoxia particularly detrimental to metabolically demanding neural tissue. In our data, bifurcation velocity (β = −0.313) was found inversely associated with CNFL, reinforcing the importance of hemodynamic dysfunction. The positive association between corneal sensitivity and CNFL (central β = 0.278, temporal β = 0.195, nasal β = 0.135) underlines again the structure–function relationship in diabetic neuropathy. Previous studies report correlation coefficients of 0.81–0.87 between CNFL and peripheral nerve conduction velocity [29], while CNFL demonstrates 85% sensitivity and 84% specificity for early neuropathy detection [11], substantially exceeding traditional screening methods. Lastly, we explored non-linear effects on CNLF using polynomial regression. The model showed a strong association with CNFL (R^2^ = 0.65). The interpretation of the model is, given the limited sample size, speculative but may serve as a hypothesis for further studies. The cubic and quadratic term may represent double (lower-bound and upper-bound) ceiling effects of CVD’s effect on CNFL, and the linear term could estimate baseline perfusion deficits of diabetic microvasculature. Corneal epithelial thickness correlates with neurotrophic factor production, creating a protective buffer against ischemic injury [30,31]. This formula may capture how neural health depends on both vascular supply and tissue capacity to generate protective factors, aligning with recent evidence showing epithelial changes as early manifestations of diabetic neuropathy [31]. Small additional vascular compromise near inflection points precipitates disproportionate neural damage.

Our results align with known pathophysiological mechanisms: carotid atherosclerosis and increased intima-media thickness indicate endothelial dysfunction of both macro- and micro-circulation. Impaired nitric oxide bioavailability reduces vasodilatory capacity, while arterial stiffening increases hemodynamic shear stress, both compromising endoneurial blood flow. In turn, this drives hypoxia in peripheral nerves, particularly vulnerable due to their elevated metabolic demands and limited collateral circulation. The fact that HbA1c was not selected as a predictor in our cohort despite a median diabetes duration of 10 years suggests that in established disease, cumulative vascular injury can supersede current glycemic control. This aligns with longitudinal studies showing accelerated small-fiber neuropathy progression relative to metabolic control in long-standing diabetes [32,33]. The comparison with healthy control groups from the literature (normative benchmarking) shows that despite modest vascular compromise, the neuropathy was severe in the majority of our patients (z-score −1.91). This may indicate that subclinical endothelial dysfunction leads to neuropathy prior to the injury of the arteries. The comparison with the literature is coherent with the vascular–neural associations identified in the above-presented regression models.

The limitations of our study include the moderate sample size (n = 30), which may miss smaller associations. Penalized regression with patient-level random effects addressed overfitting concerns, but external validation in larger independent cohorts remains essential. The cross-sectional design precludes determination of whether vascular changes precede, parallel, or follow corneal nerve fiber reduction; prospective longitudinal studies with serial CNFL and vascular assessments over 3–5 years are essential to establish temporal relationships, predictive validity, and causal inference. The absence of local controls was partially addressed through normative benchmarking (n = 343), though direct comparison would strengthen interpretation. Geographic clustering at two affiliated clinics in Romania may limit generalization. While lacking cohort-specific reproducibility metrics, we used validated automated CNFL software (published ICC = 0.87–0.94) and standardized vascular protocols by a single operator. Unmeasured confounders (vitamin B12, thyroid function, autoimmune conditions) may influence findings. The polynomial regression should be considered only exploratory before external validation on a large-scale cohort.

## 5. Conclusions

Our findings demonstrate that vascular injury is strongly associated with neural damage in diabetes, with carotid disease exerting a greater impact than metabolic parameters. The consistent lack of association between HbA1c levels and neural complications across our analyses suggests that the underlying pathophysiological mechanisms evolve with disease progression, whereby cumulative vascular injury surpasses glycemic control as the primary determinant in long-standing diabetes. These results challenge the prevailing metabolic-centered paradigm of diabetic management and underscore the importance of prioritizing vascular assessment in patients with established disease.

## Figures and Tables

**Figure 1 jcm-14-08411-f001:**
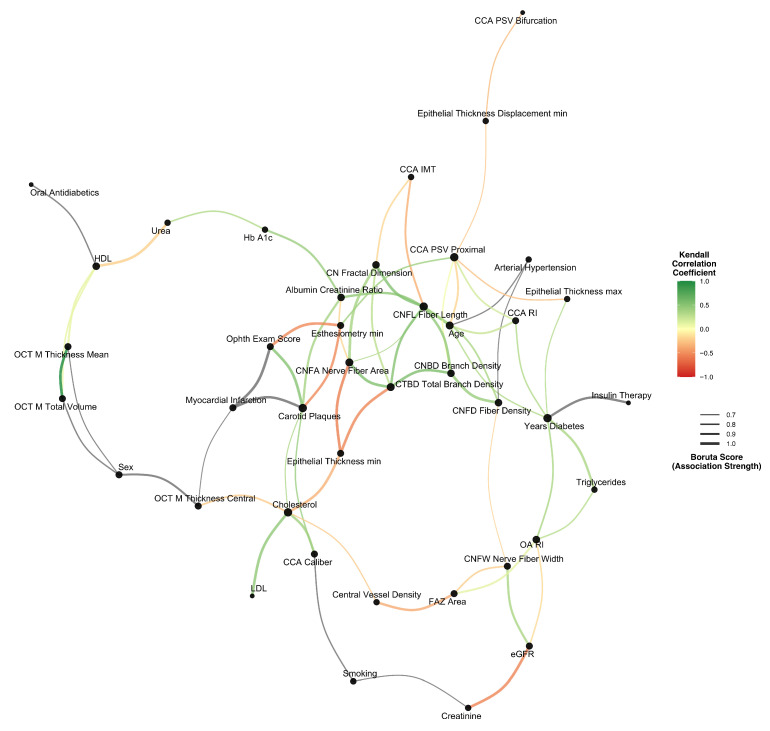
Network visualization of multivariate interactions. Edge width indicates the strength of multivariate interaction. Edge color displays the Kendall correlation coefficient.

**Figure 2 jcm-14-08411-f002:**
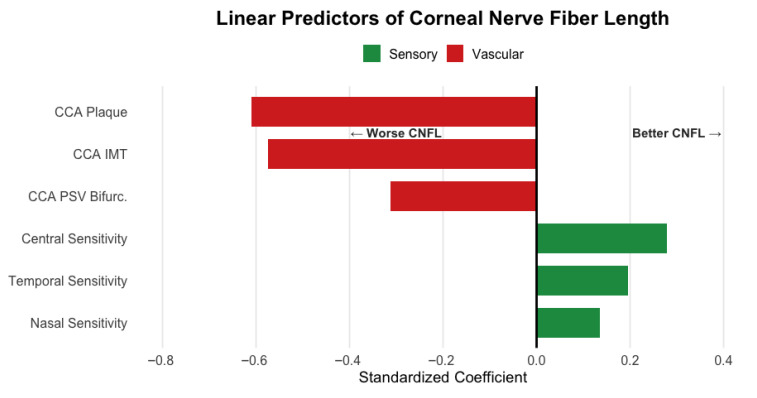
Visual summary of key findings. Vascular parameters (red) show negative associations while sensory parameters (green) show positive associations with CNFL.

**Figure 3 jcm-14-08411-f003:**
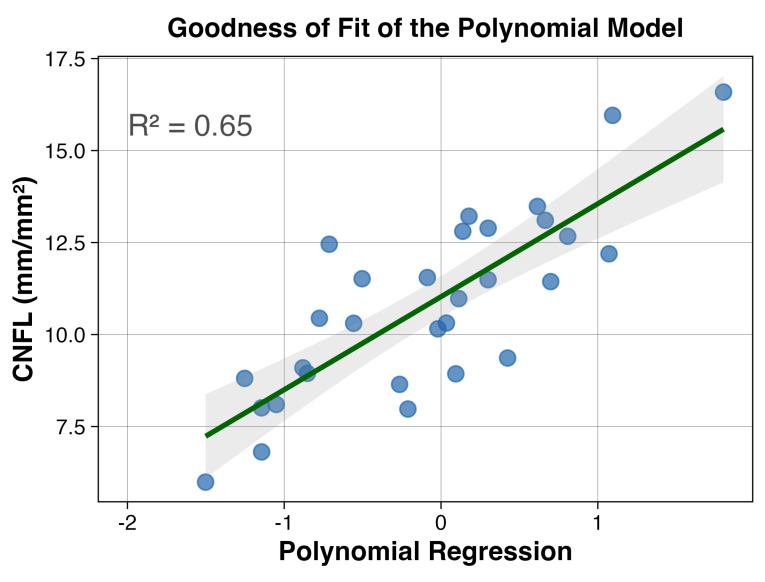
Goodness of fit for the polynomial transformation model. The polynomial transformation achieves R^2^ = 0.65, demonstrating strong predictive capability for CNFL estimation. A result of the formula below 1 indicates CNFL < 12.

**Table 1 jcm-14-08411-t001:** Descriptive statistics.

	25th Perc.	Median	75th Perc.
Age (years)	55.00	59.50	68.00
Years with diabetes	6.25	10.00	16.75
HbA1c (%)	6.53	7.05	7.40
HDL (mg/dL)	36.45	43.00	53.00
LDL (mg/dL)	104.25	123.00	144.05
Total cholesterol (mg/dL)	176.95	208.00	239.50
Triglycerides (mg/dL)	124.25	148.00	167.75
Urea (mg/dL)	30.25	37.20	45.88
Creatinine (mg/dL)	0.74	0.86	0.97
Albumin/creatinine ratio	4.64	7.45	15.07
eGFR (mL/min/1.73 m^2^)	73.42	89.00	97.07
CCA caliber (mm)	7.10	7.65	7.95
CCA IMT (mm)	0.60	0.70	0.78
CCA RI	0.70	0.74	0.77
CCA PSV proximal (cm/s)	71.75	86.00	97.50
CCA PSV bifurcation (cm/s)	68.25	78.00	87.75
OA PSV (cm/s)	38.25	41.50	46.75
OA RI	0.73	0.77	0.81
Esthesiometry min	2.25	3.00	4.00
Corneal thickness min (μm)	535.25	555.50	575.75
Epithelial thickness min (μm)	42.25	45.00	48.75
Epithelial thickness max (μm)	56.25	60.00	63.75
Central corneal thickness (mm)	0.54	0.56	0.58
Epithelial thickness displacement min (μm)	1.80	2.20	3.20
CNFD (fibers/mm^2^)	12.50	18.75	25.00
CNBD (branches/mm^2^)	12.50	18.75	25.00
CTBD (branches/mm^2^)	25.00	31.25	48.43
CNFA (mm^2^/mm^2^)	0.004	0.005	0.007
CNFW (mm)	0.02	0.02	0.02
CN fractal dimension	1.45	1.47	1.48
OCT-M thickness mean (μm)	267.38	274.55	280.17
OCT-M thickness central (μm)	181.50	204.00	218.00
OCT-M total volume (mm^3^)	7.56	7.76	7.92
FAZ area (μm^2^)	239.71	331.12	387.33
Central vessel density (%)	16.56	19.05	24.30

Abbreviations: CCA, common carotid artery; IMT, intima-media thickness; RI, resistivity index; PSV, peak systolic velocity; OA, ophthalmic artery; CNFD, corneal nerve fiber density; CNBD, corneal nerve branch density; CTBD, corneal total branch density; CNFA, corneal nerve fiber area; CNFW, corneal nerve fiber width; OCT, optical coherence tomography; OCT-M, macular optical coherence tomography; FAZ, foveal avascular zone.

**Table 2 jcm-14-08411-t002:** Linear, non-parametric and non-linear correlations.

	Pearson	Kendall’s Tau	Distance Corr.
CNFrD	0.6 [0.31–0.79], *p* < 0.001	0.53, *p* < 0.001	0.68, *p* < 0.001
CTBD	0.59 [0.3–0.79], *p* < 0.001	0.52, *p* < 0.001	0.61, *p* = 0.001
CNBD	0.59 [0.29–0.78], *p* < 0.001	0.46, *p* < 0.001	0.6, *p* < 0.001
CNFA	0.47 [0.13–0.71], *p* = 0.009	0.37, *p* = 0.005	0.49, *p* = 0.016
CNFD	0.42 [0.07–0.68], *p* = 0.021	0.35, *p* = 0.012	0.48, *p* = 0.014
CCA IMT	−0.47 [−0.71–0.13], *p* = 0.009	−0.35, *p* = 0.011	0.54, *p* = 0.007
min. Epi. Thick.	−0.4 [−0.66–0.05], *p* = 0.029	−0.3, *p* = 0.022	0.43, *p* = 0.054

Abbreviations: CNFrD, CN fractal dimension; CTBD, corneal total branch density; CNBD, CN branch density; CNFA, CN fiber area; CNFD, CN fiber density; CCA IMT, common carotid artery intima-media thickness; min. Epi. Thick., minimum epithelial thickness.

**Table 3 jcm-14-08411-t003:** Sensitivity analysis results for model stability.

Analysis	Variables Selected	Key Finding	Distance Corr.
Lambda ± 5	6 (stable)	All 6 predictors retained	0.68, *p* < 0.001
Lambda ± 10	5–7	Nasal sensitivity unstable	0.61, *p* = 0.001
Leave-one-patient-out	5–6	Core 5 predictors always selected	0.6, *p* < 0.001
Bootstrap (*n* = 100)	6 (95% of samples)	Vascular predictors most stable	0.49, *p* = 0.016

## Data Availability

Deidentified data and analysis codes are available upon reasonable request. The discovery of the coefficient error emphasizes the importance of data transparency.

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
