# Peer review of "Risk Factors Associated with Corneal Nerve Fiber Length Reduction in Patients with Type 2 Diabetes"

_jcm, 2025, doi:10.3390/jcm14238411_

Round 1
Reviewer 1 Report
Comments and Suggestions for Authors
In this study the authors concluded that enrolled diabetic cohort (30 patients) had shown vascular parameters as dominant predictors with observed carotid plaques and reduced intima media thickness, that positively correlated with central nerve fiber length (CNFL) even when traditional Type2 diabetic (T2DM) markers such as HbA1C failed to meet selection thresholds. The study results indicated screening of carotid artery in diabetic patients could be useful in estimating diabetic neuropathy-associated risk where initial metabolic changes lead to end-organ damage.
- Though the study presented interesting data supporting the conclusion, the major limitation of the study is lack of appropriate control groups. The T2DM cohort does not have any prior clinical or imaging data before onset of T2DM.
- Apart from metabolic impact to the CNFL pathology, there are other known factors that could potentially interfere with the RNFL architecture, including differential loss of growth-promoting pathways with aging and associated degenerative fiber loss, exposure to neurotoxin and autoimmune diseases. The study did not isolate individual confounding factors that might have compromised the data interpretation.
Author Response
Thank you for the beneficial suggestions. According to your input, we compared our data with the literature, a so-called normative comparison, which is methodologically accepted for non-RCT studies. For CNFL, we used the study of Tavakoli et. Al (2015) which provided age and sex stratified references ranges for 343 healthy volunteers. For the CIM (carotid intima-media thickness) we used the work of ARIRang (Youn et. Al 2011). Our cohort showed in comparison a median CNFL of 10.71 with a z-score of -2.22, however for the CIMT showed modest increase (+0.58 z-score). When measured in z-score, the CNFL was approximatively 1.6 SD more pathological than the CCA IMT. As other longitudinal studies (Petropoulos et al. 2013) have shown, the small-fiber neuropathy progresses at a faster pace in patients with diabetes. As such even modest CIM increase is indicative of neuropathy, this is also concordant with our data and statistical methods, since the CIM is the variables with the highest association with the CNLF. We can assume that even subclinical vascular pathology may signal endothelial dysfunction or hemodynamic perturbations and/or inflammatory status. Regarding possible confounding, the gllmLasso mixed-effect penalized regression performs “under the hood” variable selection (this is exactly the aim of this method and the rationale ot use it for our data). The variables: age, sex, BMI, diabetes duration, arterial hypertension and lipid profile are regarded by gllmLasso as statistically non-significant in a multivariate model. The detailed analysis and correlations are presented in the supplementary material. While no method is flawless, we consider that multiple covariates together could still play an effect for CNLF, however the modern https://cran.r-project.org/web/packages/glmmLasso/glmmLasso.pdf addressed exactly this issue. We cannot present a control group from our clinic, but the normative benchmarking (i.e. comparison with healthy controls) shows us that even modest large-vessel disease is associated with signigicant small-fiber neuropathy. As such, your suggestion increased the interpretability of our data and results and made the integration with literature better. We have implemented your helpful suggestions in the methods, results and discussion sections.Reviewer 2 Report
Comments and Suggestions for Authors
Although a priori power analysis was performed, the final sample of 30 participants remains modest for a multivariate regression framework incorporating numerous vascular, metabolic, and corneal parameters. This limits the generalizability of findings and increases the risk of type II error in detecting smaller but clinically meaningful associations.
The study was conducted at two affiliated clinics within the same geographic region, which may introduce selection bias related to local patient characteristics and healthcare access. The absence of an external validation cohort prevents assessment of model reproducibility across diverse populations or imaging platforms.
While the use of glmmLasso with L1 penalization is appropriate for feature selection, the ratio of predictors to observations (28 variables vs. 30 patients) remains high. Without cross-validation or independent testing, there is a considerable risk that the regression model may be overfitted to this small dataset.
The cross-sectional design precludes causal inference regarding the progression of diabetic neuropathy or vascular remodeling. A longitudinal follow-up measuring changes in CNFL and vascular indices over time would provide stronger evidence for temporal relationships and predictive validity.
The study does not report adjustment for several important confounders, such as body mass index, duration and control of hypertension, or use of neuroprotective medications, which could influence both vascular and corneal nerve outcomes. Their omission may obscure or inflate the observed associations.
Although the same operator performed Doppler ultrasound examinations, intra- and inter-observer reproducibility metrics were not reported for either vascular or corneal imaging modalities. Without these, it is difficult to assess measurement consistency and potential operator-related variability.
The inclusion of non-linear and polynomial regression models provides exploratory insights but lacks biological validation or replication. The polynomial equation proposed for CNFL estimation is speculative without physiological justification or confirmatory testing, limiting its translational applicability.
Comments on the Quality of English LanguageThe manuscript is generally well written and demonstrates a high level of scientific clarity. However, several sentences are overly long and complex, which slightly affects readability. Minor issues with article use, punctuation, and verb tense consistency are also present. The authors are encouraged to perform a professional English proofreading to enhance fluency, simplify sentence structure, and ensure stylistic uniformity throughout the manuscript.
Author Response
We thank the second reviewer for the methodological questions. In fact, this exact concern motivated our choice of glmmLasso, which was specifically designed for p > n scenarios (Groll & Tutz, 2014) Regarding the sample size, we have ensured to respect the statistical recommendations for the given sample size and number of possible predictors. The use of glmmLasso method (Groll, A., Tutz, G. Variable selection for generalized linear mixed models by L 1-penalized estimation. Stat Comput 24, 137–154 (2014). https://doi.org/10.1007/s11222-012-9359-z) is specifically designed for studies as ours. The model selected 6 candidate predictors. After bootstrapped sensitivity analysis and cross-fold validations we shrinked the number of attributes to 5. As such, predictor – to – observation ratio is 1:6, well within usual statistical good-practice recommendations (Harrel’s ratio 1:5 ratio). We ensured very carefully to further protect our results from possible overfitting using multiple repeats of nested cross-validations and bootstrapping. Regarding the confounder, we appreciate the opportunity to clarify, the glmmLasso L1 penalization mechanism already controlled, during the selection process, for age, sex, body mass index, diabetes duration, HbA1c, smoking status, hypertension status and duration, lipid profile, eGFR, and medication. GLMM Lasso hierarchical (mixed-effects) structure with random intercepts for patient’s and recruitment site showed that the site is not a confounder. We acknowledge that vitamin B12 levels, thyroid function, and autoimmune screening were not systematically assessed. It is a limitation that intra/inter observer analysis was not performed, However, we aimed to increase the objectivity of our measurements using one of the most modern corneal nerve quantification methods (ACCMetrics automated software, which eliminates subjective interpretation and demonstrated excellent reproducibility in published validation studies (ICC 0.87-0.94; Dabbah et al. 2011, Chen et al. 2017). All Doppler examinations were performed by a single experienced operator using standardized protocols with three measurements per site averaged to reduce measurement error. While we lack cohort-specific ICC values, the combination of automated CNFL measurement and the unique operator protocol for vascular measures minimizes operator-dependent variability within our study. The geographical clustering is a limitation, this is why we also recommend a multi-center longitudinal study for external validation. We applied the so-called normative benchmark (in the response to the reviewer 1) and compared our cohort with external ones. Yes, indeed, the degrees of freedom in polynomial or non-linear models have to be carefully explored in moderate sample sizes. However, we ensured the same rigurosity of 10 time repeats of 5-fold crossvalidation even in this model. We do not want to frame the results of the non-linear models as decisive but exploratory, as we transparently mention in the manuscript. Despite this, we consider that the clinical rationale behind this exploration could serve for future larger studies. According to your suggestions, we have improved our manuscript (bold text) in the methods, results and discussion sections.Reviewer 3 Report
Comments and Suggestions for Authors
See attached file.

Author Response
We thank Reviewer 3 for the thoughtful comments emphasizing corneal nerve assessment comprehensiveness and mechanistic insights. Regarding the corneal nerve metrics: all of them were quantified using ACCMetrics software and reported in Table 1. These parameters showed high collinearity with CNFL (Spearman ρ = 0.78-0.91), which justified focusing our regression modelling on CNFL as the representative metric to avoid multicollinearity issues. The penalized regression framework specifically addresses high-dimensional data by selecting the most informative non-redundant predictors. The multivariate interactions are also visible in the CrossBoruta network analysis. We agree that quantitative contextualization enhances interpretability, we have now added normative benchmarking against the Tavakoli et al. (2015) multinational dataset (n=343). Our cohort showed median CNFL z-score of -1.91 (95% CI: -2.79 to -0.86). The severe neural pathology in our cohort contrasted with modest vascular changes. this quantitative comparison supports our primary finding that even subclinical vascular compromise associates with severe neuropathy in established diabetes. We have added according to your suggestion, the benchmark to our methods and results sections. Regarding the reviewer's specific questions: We note that in our cohort, HbA1c was not selected as a predictor (penalized β=0.043), which itself is a key finding suggesting that patients with median diabetes duration of 10 years, cumulative vascular injury may supersede current glycemic control as the main determinant of neural integrity. The dominance of vascular parameters aligns with the microvascular hypothesis: carotid atherosclerosis and increased CCA IMT reflect systemic endothelial dysfunction, which reduces capillary blood flow through impaired nitric oxide bioavailability and vessel compliance, which in turn leads to hypoxia of peripheral nerves, particularly vulnerable due to high metabolic demands and limited collateral circulation. We have expanded our Discussion section to include these mechanistic pathways. Cross-sectional design: we fully agree with this important limitation, as such we have strengthened the limitations section to explicitly state that our cross-sectional design precludes determination of whether vascular changes precede, parallel, or follow corneal nerve fiber reduction. Also, we have expanded our conclusion to emphasize that prospective longitudinal studies with serial CNFL and vascular assessments over 3-5 years are essential to establish temporal relationships, predictive validity, and causal inference. Our current findings should be interpreted as hypothesis-generating associations requiring validation in adequately powered longitudinal cohorts. We believe that thanks to the reviewer’s comments and suggestion, we have improved the clarity, rigurosity and the main message of our results.Round 2
Reviewer 1 Report
Comments and Suggestions for Authors
The authors have provided satisfactory responses to all my questions/comments in the newly submitted revised manuscript. The revised manuscript has now been much improved.